



# TChem-atm (v2.0.0): Scalable Performance-Portable Multiphase Atmospheric Chemistry

Oscar H. Díaz-Ibarra[1], Samuel G. Frederick[2], Jeffrey H. Curtis[2], Zachary D'Aquino[2], Peter A. Bosler[1], Lekha Patel[1], Cosmin Safta[3], Matthew West[4], and Nicole Riemer[2]

[1]Center for Computing Research, Sandia National Laboratories, Albuquerque, NM, USA
[2]Department of Climate, Meteorology, and Atmospheric Sciences University of Illinois Urbana-Champaign, Urbana, IL, USA
[3]Data Sciences and Computing, Sandia National Laboratories, Livermore, CA, USA
[4]Department of Mechanical Science and Engineering, University of Illinois Urbana-Champaign, Urbana, IL, USA

**Correspondence:** Oscar H. Díaz-Ibarra (odiazib@sandia.gov) and Nicole Riemer (nriemer@illinois.edu)

**Abstract.** We present TChem-atm, a performance-portable approach that enables efficient simulation of chemically detailed and multiphase atmospheric chemistry on modern heterogeneous computing architectures. Unlike previous efforts that rely on architecture-specific code or focus exclusively on gas-phase chemistry, TChem-atm supports fully coupled gas–aerosol systems with execution across CPUs, NVIDIA GPUs, and AMD GPUs through the Kokkos programming model. It integrates the

flexible multiphase capabilities of the Community Atmospheric Model Chemistry Package (CAMP) with the high-performance kinetic routines of TChem, and includes automatic Jacobian construction with support for a range of stiff ODE solvers. We demonstrate TChem-atm's integration into the particle-resolved aerosol model PartMC and validate its accuracy against the existing PartMC–CAMP implementation, showing agreement within solver tolerances. Performance benchmarks reveal substantial speedups on GPU platforms, particularly for large particle populations, with consistent results across hardware back-

ends. By enabling chemically detailed, multiphase simulations with true performance portability and host-model flexibility, TChem-atm provides a new foundation for next-generation atmospheric models.

## 1 Introduction

Understanding the chemical evolution of the atmosphere is central to predicting air quality, climate forcing, and ecosystem impacts. Atmospheric models must capture complex interactions between gas-phase reactions, aerosol microphysics, and mul-

tiphase chemistry—often across large spatial and temporal scales. As chemical mechanisms become more detailed and models increase in resolution, the computational cost of solving the associated systems of equations grows rapidly. To address this challenge, flexible and scalable methods are needed that can integrate complex chemistry into models while taking full advantage of modern computing architectures.

Simulating detailed gas-phase mechanisms, multiphase processes, and aerosol microphysics across fine spatial and tempo-

ral scales often requires solving thousands of coupled, numerically stiff ordinary differential equations (ODEs) at each grid point. This makes chemical solvers one of the most computationally intensive components of atmospheric models. Graphics



Processing Units (GPUs) offer a massively parallel architecture that can accelerate these calculations significantly, enabling more detailed and efficient simulations.

This acceleration is critical across a wide range of applications. In earth system models such as E3SM (Energy Exascale Earth System Model), interactive chemistry and aerosols are essential for representing radiative forcing and cloud feedbacks over long timescales (Golaz et al., 2019; Rasch et al., 2019). In global chemical transport models like GEOS-Chem, GPU acceleration enables higher spatial resolution and real-time data assimilation (Bey et al., 2001; Eastham et al., 2018). Operational air quality models such as CMAQ must meet tight runtime constraints while still representing complex chemical and physical processes (Byun and Schere, 2006; Appel et al., 2021). Across these domains, reducing the computational burden of atmospheric chemistry is essential for improving both fidelity and performance.

Recent work has demonstrated the potential of GPU acceleration for atmospheric models. For example, Ruiz et al. (2024) achieved a 35× speed-up by distributing the CAMP solver load across GPU threads, and Alvanos and Christoudias (2017) reported kernel-level speed-ups of up to 20.4× for CUDA-based kinetic integration in EMAC. Cao et al. (2023) showed more than 1000× speed-ups for GPU-accelerated advection in CAMx using hybrid GPU–MPI strategies, while Quevedo et al. (2025) demonstrated that GPU-enabled CMAQ simulations could cut runtimes nearly in half. Similarly, Sun et al. (2018) highlighted the importance of memory layout and GPU-aware optimization in achieving efficient Rosenbrock solver performance in CAM4-Chem.

While these studies underscore the promise of GPU-accelerated atmospheric chemistry, most rely on architecture-specific APIs (e.g., CUDA) and hand-optimized kernels tightly coupled to the host model. This approach limits code portability, increases integration complexity, and hinders broader adoption of GPU acceleration across diverse computing environments.

To address these limitations, performance-portable approaches such as TChem have emerged, leveraging libraries like Kokkos to abstract architecture-specific details (Kim et al., 2023). TChem provides a scalable and efficient backend for evaluating gas-phase and surface reaction kinetics, thermodynamic properties, and Jacobians via automatic differentiation. It is designed for portability across heterogeneous architectures, enabling the same codebase to run efficiently on CPUs, NVIDIA GPUs, and AMD GPUs. However, TChem was originally developed for gas-phase chemistry and did not natively support multiphase processes, limiting its application in comprehensive atmospheric chemistry models.

In parallel, we developed the Chemistry Across Multiple Phases (CAMP) library as a runtime-configurable tool for simulating gas- and aerosol-phase chemistry in models with varying aerosol representations (Dawson et al., 2022). CAMP decouples chemical logic from the host model's aerosol structure and supports modular chemical mechanism configuration and solver abstraction. However, while CAMP is flexible in mechanism design, its reliance on host-side solvers limited its scalability and portability to new hardware architectures.

Recognizing their complementary capabilities, the present work combines CAMP's support for multiphase chemistry with TChem's performance-portable computational backend. While both CAMP and TChem provide flexibility in chemical mechanism configuration, CAMP offers a structure tailored for gas–aerosol interactions, whereas TChem enables efficient execution on heterogeneous architectures via the Kokkos programming model. The resulting merged library supports simulations of gas- and aerosol-phase chemistry across CPUs and GPUs, and provides interoperability with multiple ODE solvers, including





Tines (Kim and Diaz-Ibarra, 2021) and SUNDIALS (Balos et al., 2021; Gardner et al., 2022; Hindmarsh et al., 2005, 2025). Since TChem is primarily used for gas-phase chemistry, particularly in combustion applications, we decided to name the specialized version of TChem for atmospheric chemistry TChem-atm. It is important to note that TChem-atm is a standalone code, distinct from TChem, and has its own dedicated GitHub repository.

This paper presents the first application of this integrated library to atmospheric modeling, with a focus on enabling efficient aerosol–chemistry interactions in high-resolution simulations. The ability to interchange solvers across platforms without reimplementing solver logic streamlines deployment and optimization. By supporting scalable chemistry calculations on diverse architectures, this work lays the foundation for next-generation weather, climate, and air quality models, and represents a concrete step toward the generalized aerosol/chemistry interface advocated by Hodzic et al. (2023). We note that while our current implementation succeeds as a first step toward performance portability, individual applications could be individually tuned for even greater performance. Such tunings would depend on the chemistry mechanisms present in the application, the choice of solver, and the specific computational architecture. Here, we introduce the capability to flexibly define atmospheric chemistry mechanisms for a variety of applications in a common computational environment, capable of running on a variety of advanced computing architectures; optimized tunings for specific applications, solvers, and architectures is a subject marked for future work.

## 2 Methods

TChem-atm is a performance-portable chemistry library designed to support multiphase atmospheric chemistry across a range of computing architectures. Figure 1 presents a high-level overview of the TChem-atm library and outlines the structure of this section. The workflow begins with user-defined YAML input files that specify gas-phase chemical mechanisms and aerosol processes (Section 2.1). These inputs are parsed into structured data objects for the gas and aerosol phases (Section 2.2), which are synchronized to device memory via dual Kokkos views. TChem-atm then constructs source terms for gas-phase chemistry (Section 2.4) and for aerosol-phase processes using a flexible abstraction that supports sectional and particle-resolved representations (Sections 2.6 and 2.5). Performance portability is achieved through the Kokkos programming model (Section 2.7), which enables execution on CPUs as well as NVIDIA and AMD GPUs. If needed, Jacobian matrices are computed (Section 2.8) and passed to one of several supported ODE solvers (Section 2.9). Our testing approach is described in Section 2.10. Finally, we describe the integration of TChem-atm with PartMC, which enables particle-resolved aerosol simulations using batched, scalable multiphase chemistry (Section 2.11). The following subsections provide implementation details for each component of the implementation.

### 2.1 Chemical Mechanism Configuration

To support flexibility and ease of use, both CAMP and TChem-atm adopt human-readable configuration systems based on JSON or YAML for defining chemical mechanisms and model parameters (Dawson et al., 2022; Kim et al., 2023). This design enables users to specify complex reaction mechanisms, thermodynamic properties, and aerosol interaction parameters



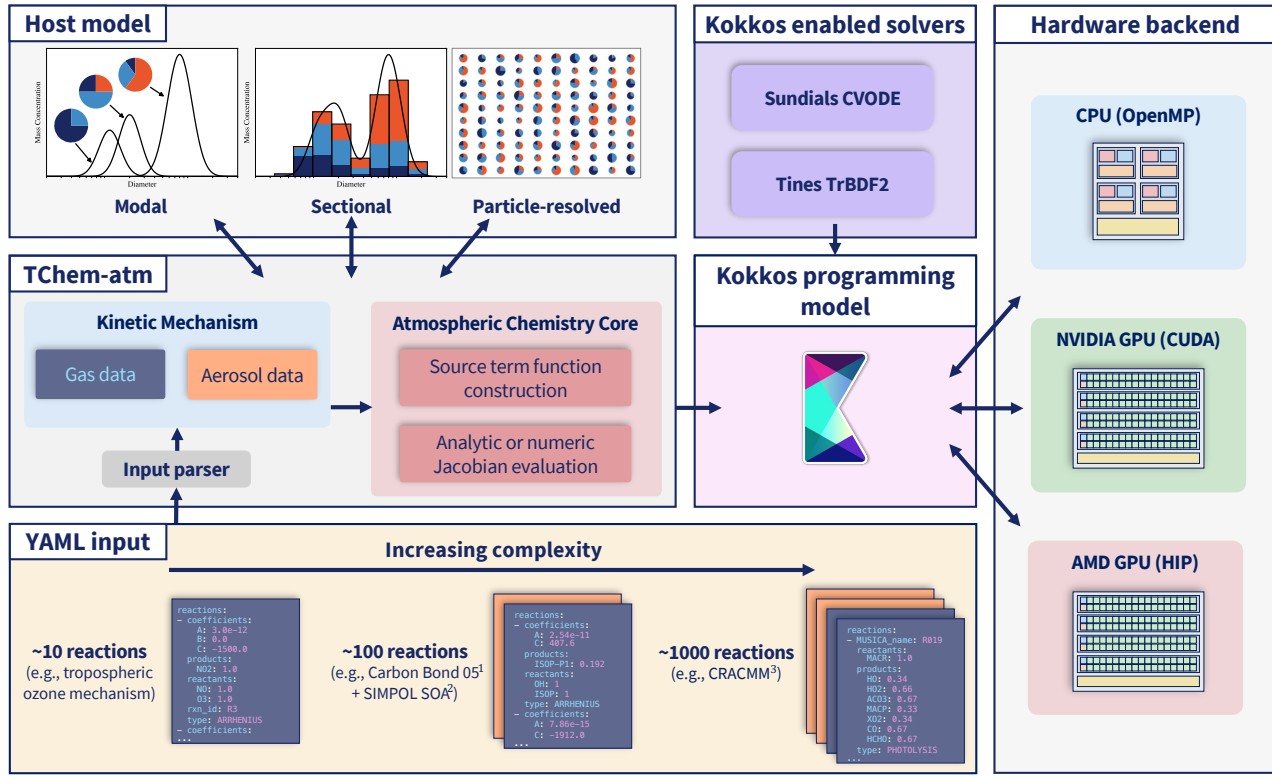

**Figure 1.** Diagram of TChem-atm highlighting the customizability of the chemical mechanism and interoperability with numerous host model aerosol treatments. Further customization is supported for the selection of various Kokkos-enabled solvers, which allow performance portable computing across numerous platforms. [1]Carbon Bond 05: Yarwood et al. (2005); [2]SIMPOL: Pankow and Asher (2008); [3]CRACMM: Pye et al. (2023).





```
environmental_conditions:
  pressure:
    initial_value: [P1, ..., PN]
    units: Pa
  temperature:
    initial_value: [T1, ..., TN]
    units: K
initial_state:
  A:
    initial_value: [A1, .., AN]
    units: mol m-3
  B:
    initial_value: [B1, .., BN]
    units: mol m-3
  C:
    initial_value: [C1, .., CN]
    units: mol m-3
reactions:
- coefficients:
    A: 1476.0
    Ea: 5.5e-21
    B: 150.0
    E: 0.15
  products:
    C: 1.0
  reactants:
    A: 1.0
    B: 1.0
  type: ARRHENIUS
species:
- description: A
  name: A
- description: B
  name: B
- description: C
  name: C
```

**Figure 2.** Example YAML input specifying a simple gas-phase mechanism consisting of a single reaction with three species $(A + B \rightarrow C)$ used in simulations with $N$ independent computational cells, each solving a separate instance of the ODE system.

at runtime, without modifying the core codebase. Figure 2 shows an example gas-phase mechanism in YAML format for a

90  simple system involving one reaction and three species $(A + B \rightarrow C)$. This configuration is used to simulate the same chemical system independently across $N$ computational cells, where each cell represents a distinct instance of the ODE system such as grid points in a host model or batches in a unit test.

The structured and editable format of these configuration files, along with the availability of validation tools, makes them accessible to a broad range of users—including those without specialized programming experience. In this work, we retain

95  CAMP's approach for specifying multiphase chemistry processes such as gas–aerosol partitioning and condensed-phase reactions, and integrate it into the TChem-atm infrastructure. This unification allows researchers to flexibly configure chemically detailed multiphase systems in a portable, performance-optimized environment suited for both comprehensive atmospheric models and standalone simulations (e.g., chamber or flow-tube studies).

### 2.2  Aerosol and Gas Kinetic Model Data

100  TChem-atm organizes the reaction mechanism information for gas and aerosol phases into two structured data objects as shown in Figure 3. The gas phase data is stored in the Kinetic Model Constant Data (kmcd) object, while the aerosol phase data is stored in the Aerosol Model Constant Data (amcd) object.

Constructing these data objects involves two steps. First, the reaction mechanism is parsed and the relevant kinetic or thermodynamic information is stored in intermediate Kinetic Model and Aerosol Model data objects. These data are stored in dual

105  Kokkos views, which allow synchronization of data between host and device memory. The data are further categorized and



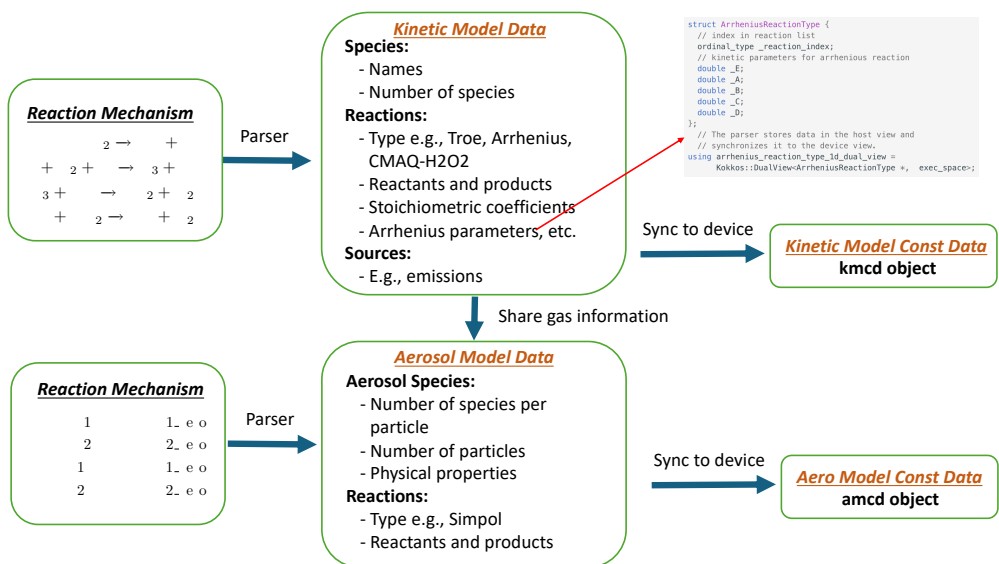

**Figure 3.** TChem-atm's internal representation of the chemical mechanism is the kinetic model object.

organized to optimize device-side kernel computation and facilitate postprocessing. For instance, for reactions following the Arrhenius form, TChem-atm stores both the reaction type and the associated parameters in separate dual views, as presented in Figure 3.

Second, TChem-atm extracts only the data required by the computational kernels and synchronizes this subset to the device. The resulting constant data object—either `kmcd` or `amcd`—is then passed to the computation kernel. As a constant object, it cannot be modified during computation.

### 2.3 System state vector and governing equations

We define the system state vector $\boldsymbol{\eta}$ as the concatenation of the gas-phase state vector $\boldsymbol{g}$ and aerosol-phase chemical state vector $\boldsymbol{\mu}$ such that $\boldsymbol{\eta} = [\boldsymbol{g}, \boldsymbol{\mu}]$. The gas-phase state is represented by mixing ratios of $K$ chemical species $\boldsymbol{g} = [g_1, g_2, \ldots, g_K]$, while the aerosol phase is described by a set of $N$ aerosol-phase vectors $\boldsymbol{\mu} = [\boldsymbol{\mu_1}, \boldsymbol{\mu_2}, \ldots, \boldsymbol{\mu_N}]$. Each aerosol-phase vector $\boldsymbol{\mu_i}$ can be thought of as a "computational particle" $i$ and contains the mass of $A$ tracked species, $\boldsymbol{\mu_i} = [\mu_i^1, \mu_i^2, \ldots, \mu_i^A]$.

In this section, we assume a particle-resolved context for illustrative purposes, where each $\boldsymbol{\mu_i}$ corresponds to a single particle. However, TChem-atm can also be applied to sectional and modal models, where the computational particles are then mapped to sections and representative particles in a mode. A detailed discussion of this mapping is provided in Section 2.6.

In expanded form, the system state vector $\boldsymbol{\eta}$ is

$$\boldsymbol{\eta} = \Big[ \underbrace{g_1, g_2, \ldots, g_K}_{\text{gas}}, \underbrace{\mu_1^1, \mu_1^2, \ldots, \mu_1^A}_{\text{particle 1}}, \underbrace{\mu_2^1, \mu_2^2, \ldots, \mu_2^A}_{\text{particle 2}}, \ldots, \underbrace{\mu_N^1, \mu_N^2, \ldots, \mu_N^A}_{\text{particle } N} \Big]. \tag{1}$$





In addition to the chemical state vector $\boldsymbol{\eta}$, TChem-atm also requires the number concentration associated with each computational particle, denoted $n_i$ for particle $i$. This number concentration is not part of the state vector because it is not modified within TChem-atm. Instead, it is passed in by the host model and may evolve independently due to physical processes such as coagulation, nucleation, or sedimentation. Nonetheless, $n_i$ plays a critical role in the calculation of aerosol chemical source terms and is required for constructing the coupled gas–aerosol source terms described below.

The governing equations for the gas-phase and aerosol-phase components of the state vector are given by:

$$\frac{dg_k}{dt} = F_{\mathrm{gg},k} + F_{\mathrm{ga},k}, \qquad\qquad k = 1,\dots,K \tag{2}$$

$$\frac{d\mu_i^a}{dt} = F_{\mathrm{ag},i}^a, \qquad\qquad i = 1,\dots,N, \quad a = 1,\dots,A \tag{3}$$

The source term $F_{\mathrm{gg},k}$ accounts for gas-phase chemical reactions that produce or consume species $k$, as specified by the gas-phase chemical mechanism and further explained in Section 2.4. The term $F_{\mathrm{ga},k}$ represents gas–aerosol coupling processes, such as condensation of low-volatility species, evaporation of semi-volatile compounds, or uptake via heterogeneous reactions, and thus transfers mass between the gas phase and the aerosol phase. The corresponding aerosol-phase term, $F_{\mathrm{ag},i}^a$, describes the gain or loss of species $a$ in computational particle $i$ due to these interactions. These coupling terms ensure mass conservation across phases and encode the dynamic exchange of chemical species between gas and aerosol reservoirs and are explained in detail in Section 2.5. TChem-atm integrates the governing equations forward in time using the Tines or Sundials CVODE ODE solver.

## 2.4 Gas Chemical Mechanism Source Term Construction

The gas-phase chemical source term, $F_{\mathrm{gg},k}$ in Eq. (2), is computed via:

$$F_{\mathrm{gg},k} = \sum_{j=1}^{N_{\mathrm{react}}} \nu_{kj}\, q_j, \quad \nu_{kj} = \nu_{kj}'' - \nu_{kj}', \tag{4}$$

Here $q_j$ is the reaction rate of reaction $j$, $\nu_{kj}'$, $\nu_{kj}''$ are stoichiometric coefficients for species $k$ in reaction $j$, and $N_{\mathrm{react}}$ is the total number of gas-phase reactions. The reaction rate is given by:

$$q_j = k_{\mathrm{f},j} \prod_{k=1}^{K} g_k^{\nu_{kj}'}, \tag{5}$$

where $k_{\mathrm{f},j}$ is the reaction rate constant for reaction $j$. The functional forms for $k_{\mathrm{f},j}$ depend on reaction type, as listed in Table 1.

Currently, TChem-atm can reproduce gas chemistry for two complex reaction mechanisms: the gas chemistry of E3SM v3, i.e., the UCI chemistry system (University of California Irvine), and the Carbon Bond 2005 chemical mechanism, which is well-formulated for urban to remote tropospheric conditions (Yarwood et al., 2005). Because TChem-atm is mechanism-agnostic, it can be easily adapted to support a wide range of applications. For example, a minimal chemical mechanism can be defined for instructional use in classroom settings, while detailed mechanisms with hundreds of species and reactions can be incorporated for high-fidelity research simulations.





**Table 1.** Gas reaction types available in TChem-atm; further details can be found in the TChem-atm online documentation.

| Reaction Type | Equation | Comments |
|---|---|---|
| Arrhenius Type | $k_{\mathrm{f}} = A\exp\left(\frac{C}{T}\right)\frac{T}{D^B}(1+EP)$ | $A$, $B$, $C$, $D$, and $E$ are kinetic constants. |
| Troe Type | $k_{\mathrm{f}} = \frac{k_0[M]}{1+\frac{k_0[M]}{k_\infty}}F_c^{\left(1+\left(\frac{\log_{10}\left(\frac{k_0[M]}{k_\infty}\right)}{N}\right)^2\right)^{-1}}$ | $k_0$ and $k_\infty$ are computed as : $k_0 = k_{0_A}\exp\left(\frac{k_{0_C}}{T}\right)\left(\frac{T}{300}\right)^{k_{0_B}}$ $k_\infty = k_{\infty_A}\exp\left(\frac{k_{\infty_C}}{T}\right)\left(\frac{T}{300}\right)^{k_{\infty_B}}$ $F_c$ is a kinetic constant. |
| Custom H2O2 Type | $k_{\mathrm{f}} = A_1\exp\left(\frac{C_1}{T}\right)\left(\frac{T}{300}\right)^{B_1} + A_2\exp\left(\frac{C_2}{T}\right)\left(\frac{T}{300}\right)^{B_2}V_A$ | $V_{\mathrm{A}} = \frac{P\,N_{\mathrm{A}}\,R\times10^{12}}{T}$, where $N_{\mathrm{A}} = 6.02214179\times10^{23}$ is Avogadro's number ($\mathrm{mole}^{-1}$), and $R = 8.314472$ is the universal gas constant ($J\,\mathrm{mole}^{-1}\,K^{-1}$). |
| Custom OH_HNO3 | $k_{\mathrm{f}} = k_{\mathrm{troe}} + k_{\mathrm{arrhenius}}$ | The Carbon Bond 05 mechanism employs this reaction type and can be expressed as the sum of Arrhenius and Troe reaction types. |
| Troe-Arrhenius Ratio Type | $k_{\mathrm{f}} = \frac{k_{\mathrm{troe}}}{k_{\mathrm{arrhenius}}}$ | This reaction type is computed as the ratio between Troe (or JPL) and Arrhenius types. |

## 2.5 Aerosol Mechanism Source Term Construction

TChem-atm computes the aerosol-gas interaction source terms, $F_{\mathrm{ga},k}$ and $F_{\mathrm{ag},i}^a$, as follows. In this section, we focus on the partitioning of semi-volatile gases. For simplicity, we assume that the first $N_{\mathrm{proc}}$ entries in the gas-phase species list correspond
to partitioning species, and that each of these has a matching aerosol-phase counterpart with the same index $a$. That is, gas species $g_a$ partitions to aerosol species $\mu_i^a$ for $a = 1, \ldots, N_{\mathrm{proc}}$.

$$F_{\mathrm{ga},a} = -\sum_{i=1}^{N} k_i^a(g_a - g_a^*), \qquad a = 1, \ldots, N_{\mathrm{proc}}, \tag{6}$$

$$F_{\mathrm{ga},k} = 0, \qquad k = N_{\mathrm{proc}} + 1, \ldots, K, \tag{7}$$

$$F_{\mathrm{ag},i}^a = \frac{\rho^a}{n_i}k_i^a(g_a - g_a^*), \tag{8}$$

where $k_i^a$ ($\mathrm{s}^{-1}$) is the mass transfer coefficient for species $a$ and particle $i$, $g_a^*$ ($\mathrm{mole\,mole}^{-1}$) is the equilibrium mixing ratio of species $a$, $\rho^a$ ($\mathrm{kg\,m}^{-3}$) is the mass density of gas species $a$ needed for the conversion of mixing ratio to mass concentration, and $n_i$ is the number concentration of computational particle $i$ needed to convert mass concentration to mass. While $F_{\mathrm{ga},a}$ is





the *mixing ratio flux* of species $a$ to the gas phase from all particles, $F_{\mathrm{ag},i}^a$ is the *mass flux* of species $a$ to computational particle $i$.

The mass transfer coefficient $k_i^a$ and the mass density $\rho^a$ are given by,

$$k_i^a = 4\pi r_i D_{\mathrm{g}}^a n_i f(\mathrm{Kn}_i^a, \alpha_a), \tag{9}$$

$$\rho^a = \frac{pM^a}{RT}, \tag{10}$$

where $r_i$ is the effective radius of the particle, $D_{\mathrm{g}}^a$ is the gas-phase diffusion coefficient of species $a$, and $f(\mathrm{Kn}_i^a, \alpha_a)$ is the Fuchs-Sutugin correction factor, $p$ is atmospheric pressure, $M^a$ is the molar weight of gas $a$ ($\mathrm{kg\,mole^{-1}}$), $R$ is the universal gas constant ($\mathrm{J\,mole^{-1}\,K^{-1}}$), and $T$ is temperature (K).


The effective radius $r_i$ can be derived from the particle masses $\mu_i^a$ and aerosol species densities $\rho_{\mathrm{p}}^a$ by assuming sphericity:

$$r_i = \left( \frac{3}{4\pi} \sum_{a=1}^A \frac{\mu_i^a}{\rho_{\mathrm{p}}^a} \right). \tag{11}$$

The Fuchs-Sutugin correction factor is computed as:

$$f_{\mathrm{fs}}(\mathrm{Kn}_i^a, \alpha_a) = \frac{0.75\alpha_a(1+\mathrm{Kn}_i^a)}{(\mathrm{Kn}_i^a)^2 + (1+0.283\alpha_a)\mathrm{Kn}_i^a + 0.75\alpha_a}, \tag{12}$$

where $\mathrm{Kn}_i^a = \lambda_a/r_i$ is the Knudsen number, $\lambda_a$ is the mean free path of gas $a$, and $\alpha_a$ is the mass accommodation coefficient of gas $a$, here assumed to be 0.1 for all gases.

To calculate the equilibrium mixing ratios $g^*$ we use the SIMPOL.1 method (Pankow and Asher, 2008), which parameterizes $g_a^*$ as follows:

$$g_a^* = \frac{p_a^*}{p}, \tag{13}$$

$$\log_{10} \frac{p_a^*}{p_0} = \frac{B_1^a}{T} + B_2^a + B_3^a T + B_4^a \ln(T), \tag{14}$$

where $p_a^*$ is the equilibrium vapor pressure for gas $a$ (Pa), $p_0 = 101325.0$ Pa is standard atmospheric pressure and $B_1^a$, $B_2^a$, $B_3^a$, $B_4^a$ are fitting parameters provided by SIMPOL.1.

## 2.6   Abstracting Aerosol Representations Across Models

One of CAMP's key innovations is to perform chemistry on individual computational particles, providing a unified library
that works across different aerosol representations. As previously explained, a computational particle may represent an actual particle (in particle-resolved models), a bin section (in sectional models), or a representative particle within a mode (in modal models). This approach replaces the traditional coupling of chemistry to specific aerosol schemes—which is often hard-coded and inflexible—with a more general method that simplifies the addition of new species or processes. While this idea was originally described as an "abstraction of aerosol representation" in Dawson et al. (2022), the current framing in terms of
computational particles offers a more intuitive and practical perspective, and steps toward the generalized representations envisioned in Hodzic et al. (2023).





Following this approach, TChem-atm defines a general interface for aerosol–gas chemical interactions. Rather than prescribing how aerosols must be represented, TChem-atm relies on the host model to provide two essential pieces of information for each computational particle: (1) the chemical composition (i.e., the vector of aerosol species masses), and (2) the associated

number concentration. As noted in Section 2.3, the number concentration is not part of the TChem-atm state vector, since it is not modified by the chemistry, but it plays a critical role in constructing source terms and must be supplied by the host model. Similarly, the particle diameter is required for processes such as surface-area-based uptake and size-dependent reaction rates. It is diagnosed from the particle masses and the number concentrations (assuming spherical particles). In particle-resolved models, this diameter is simply the diameter of the individual computational particle. In sectional models, it is typically taken

as the midpoint of the bin.

For modal models, however, the situation is more complex. Modal schemes typically do not pass explicit diameters to the chemistry module; instead, they compute the effects of condensation by integrating over assumed size distributions and updating moments analytically—an approach rooted in the work of Whitby and McMurry (1997). While this method is widely used, it complicates the integration with approaches like TChem-atm that expect a well-defined particle diameter for each

computational particle. This limitation is not a fundamental barrier, but addressing it would require additional work. Specifically, modal host models could be adapted to pass representative diameters (e.g., mean or characteristic values), which would improve compatibility with TChem-atm and enhance transparency and efficiency in multiphase chemistry calculations.

TChem-atm requires that each computational particle contains the full set of aerosol-phase species defined by the chemical mechanism. While individual species masses may be zero in any given computational particle, all species must be present in the

data structure. This design ensures a consistent state vector structure across computational particles and simplifies source term construction. As a result, TChem-atm does not currently support mode-specific species sets, such as those used in some modal models where particular modes are restricted to subsets of species (e.g., black carbon only, or sea salt with organic carbon only). Any such mapping or aggregation must be handled externally by the host model prior to passing data to TChem-atm.

The abstractions within TChem-atm also facilitate integration with emerging or hybrid aerosol representations, such as

adaptive sectional models, quadrature-based methods, or machine-learning-driven representations, as long as the host model can supply the required species masses, number concentrations, and diameters for each computational particle. Figure 4 illustrates how computational particles in TChem-atm correspond to different host model representations. For sectional and particle-resolved methods, there is a one-to-one mapping between computational particles tracked by TChem-atm and the corresponding host representation. For the modal approach, we display a scenario in which each mode is represented by a set of

three characteristic computational particles with diameters corresponding to the geometric mean diameter and one standard deviation above and below.

## 2.7 Performance Portability with Kokkos

To enable performance portability across diverse computing platforms, PartMC-TChem-atm leverages the Kokkos library (Edwards et al., 2014; Trott et al., 2021, 2022). Kokkos is a programming model designed to abstract parallel execution and

memory management, allowing a single implementation of numerical algorithms to run efficiently on a variety of architec-



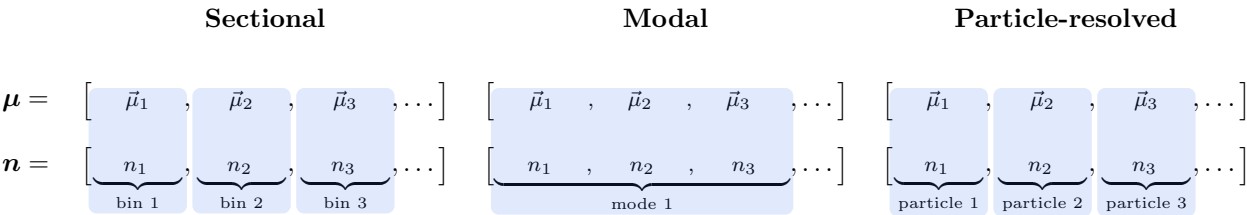

**Figure 4.** Representation of the mapping between the abstract representation of the aerosol state vector $\boldsymbol{\mu}$ in TChem-atm and the host model representation (blue filled regions). The host model is responsible for tracking the number concentration of each computational particle, contained in the $\boldsymbol{n}$ vector. Particle diameters are diagnosed from $\boldsymbol{\mu}$ and $\boldsymbol{n}$.

tures, including multi-core CPUs (OpenMP), NVIDIA GPUs (CUDA), AMD GPUs (HIP), and Intel GPUs (SYCL). Kokkos achieves this through compile-time polymorphism and execution policies that separate algorithm logic from hardware-specific optimizations. This abstraction is critical for modern atmospheric modeling, where simulations must scale across heterogeneous high-performance computing (HPC) systems.

Kokkos has been adopted in several modeling efforts to modernize physics and chemistry components for exascale computing. It underpins performance-portable implementations in the Energy Exascale Earth System Model (E3SM) atmosphere component (Taylor et al., 2023) and serves as the parallel backend for TChem (Kim et al., 2023). By integrating TChem-atm with Kokkos, PartMC-TChem-atm inherits the ability to execute detailed multiphase chemistry on both CPU and GPU architectures without modifying core algorithms.

**2.8  Software Architecture and Jacobian Computation**

TChem-atm adopts the same software design as the TChem library (Kim et al., 2023) and uses batched computations and algorithm implementations based on two primary Kokkos abstractions, `ValueType` and `DeviceType`. `ValueType` represents built-in scalar data types such as `int` and `float`, but it can also use automatic differentiation (AD) types from, for example, the Sacado package to support automatic computation of Jacobians and to facilitate differentiability for machine learning.

`DeviceType` defines a memory space (where data lives, e.g., on CPU or GPU) and an execution space (where kernels run, e.g., on CPU or GPU). Each Kokkos backend supports a specific `DeviceType` so that applications written in Kokkos, using these abstractions, can run on diverse architectures. The workflow of TChem-atm is shown in Algorithm 1. It creates two data objects for gas and aerosol chemistry and uses hierarchical parallelism to batch-evaluate source terms or initial value problems. Nested team-thread parallelism enables performance across varying system configurations.

TChem-atm automatically computes Jacobian matrices using either finite difference methods or automatic differentiation via SACADO (Phipps et al., 2020). When `ValueType` is `double`, a numerical Jacobian is computed; TINES supports forward differencing, central differencing, and Richardson extrapolation, with forward differencing used by default. For analytical





Jacobians, `ValueType` is an automatic differentiation type, and Sacado computes derivatives using the chain rule. This unified design simplifies the interface for downstream solvers.

As previously discussed, TChem-atm leverages Kokkos' team parallelism, utilizing batched computation. Specifically, multiple evaluations of nested routines to invoke a team function. The number of batches or samples is directly correlated with the number of grid points, as TChem-atm is invoked by an atmospheric model, or with model evaluations, where TChem-atm is utilized for uncertainty quantification and machine learning studies. Within the batched loop, a team routine is executed, with the primary team routine in TChem-atm being the right-hand-side routine, as shown in Algorithm 1.

For the gas component of the source term, nested parallel loops are employed to compute the kinetic constants for each reaction type (see Table 1), the rate of progress for each reaction, and the net production rate. For the aerosol component, nested parallel loops are similarly utilized for the evaluation of each aerosol process, where the number of particles or sections determines the size of the loop; currently, the SIMPOL process (Tsigaridis and Kanakidou, 2007) is supported within TChem-atm. Due to this batched design, we do not anticipate superior computational performance from TChem-atm when only a single

evaluation is conducted. However, enhanced performance is expected as the number of evaluations increases.



---

**Algorithm 1** Batched algorithm for TChem-atm computation of chemical source terms.

**Input:** chemFile, aeroFile, inputFile, number_of_batches, team_size, vector_length

**Output:** Batched RHS computation

1. Select execution space based on computer architecture

device_type ← Tines::UseThisDevice<TChem::exec_space>::type

2. Define value type for Jacobian computation (double/SACADO type for numerical/analytical Jacobian)

value_type ← double

3. Read gas kinetic information from chemFile. Then, create and sync data object for gas phase

kmd ← TChem::KineticModelData(chemFile)

kmcd ← TChem::create_KineticModelConstData<device_type>(kmd)

4. Read aerosol kinetic information from aeroFile. Then, create and sync data object for aerosol phase

amd ← TChem::AerosolModelData(aeroFile, kmd)

amcd ← TChem::create_AerosolModelConstData<device_type>(amd)

5. Define parallelism policy for TChem-atm

policy_type policy ← Kokkos::TeamPolicy(number_of_batches, team_size, vector_length)

6. Read gas and aerosol state

state ← TChem::read_state(inputFile)

7. Launch kernel to device.

**do in parallel**

**for** *each batch* **do**
    // Invoke nested routines for batch i
    **do in parallel**
    **for** *each gas reaction type* **do**
        compute_kinetic_constant(state$_i$, kmcd)
    **end**
    **do in parallel**
    **for** *each gas reaction* **do**
        compute_rate_of_progress(state$_i$, kmcd)
    **end**
    **do in parallel**
    **for** *each gas species* **do**
        compute_net_production(state$_i$, kmcd)
    **end**
    **do in parallel**
    **for** *each computational particle* **do**
        compute_particle_contribution(state$_i$, kmcd, amcd)
    **end**
**end**

---



## 2.9 ODE Solvers and Integration Workflow

TChem-atm is designed to decouple the specification of chemical processes from the numerical integration of the resulting ODE systems. In atmospheric models, gas- and aerosol-phase chemistry are typically solved independently at each grid point, often requiring the integration of stiff ODE systems over short time scales.

To support this need, TChem-atm provides interfaces to two widely used implicit solvers: TINES (Kim and Diaz-Ibarra, 2021) and SUNDIALS CVODE (Gardner et al., 2022; Hindmarsh et al., 2005; Balos et al., 2021; Hindmarsh et al., 2025), both of which implement backward differentiation formula (BDF) methods suitable for stiff systems. TINES employs a second-order Trapezoidal BDF (TrBDF2) scheme (Bank et al., 1985) with a Newton-based nonlinear solver and UTV decomposition for solving linear systems. CVODE supports both dense LU-based solvers, which require a Jacobian matrix and can use either

analytical Jacobians (computed via automatic differentiation with SACADO (Phipps et al., 2020)) or numerical Jacobians (via finite difference schemes) (Salane, 1986), and GMRES-based iterative solvers, which do not require explicit Jacobians and instead approximate the action of the Jacobian through matrix-free methods.

To enable scalable integration across many independent ODE systems, TChem-atm adopts a batched interface, where source term evaluations and Jacobian computations are executed in parallel using Kokkos' team-level parallelism. This structure is

compatible with large-scale simulations that solve thousands to millions of ODE systems simultaneously, such as grid-point chemical updates in 3D models.

Users can configure solver settings—including absolute and relative tolerances, maximum time steps, and method-specific options—via human-readable input files. This flexibility, along with modular interfaces to both solvers, allows users to tailor solver behavior to problem stiffness and computational constraints without modifying the underlying chemical mechanism or

solver logic.

The interchangeable solver design also facilitates future extensibility. Additional solvers can be integrated with minimal code changes, supporting experimentation with different numerical strategies or coupling to new host models. Together, these features make TChem-atm a flexible and performance-portable platform for integrating chemically detailed, stiff ODE systems in a wide range of atmospheric chemistry applications. Performance comparisons across solver configurations—including

dense and Jacobian-free methods on CPU and GPU architectures—are discussed in Section 3.2.

## 2.10 Testing Strategy

Development of TChem-atm is maintained via a GitHub repository. To ensure changes to the code base do not break model execution, numerous tests are run via GitHub Actions (GitHub, 2025) upon both the opening and merging of pull requests. Tests include small-scale unit testing of reaction and production rates, as well as larger-scale integration tests in which the

entire chemical mechanism including E3SM's UCI chemistry system and the Carbon Bond 05 mechanism are evaluated. Additionally, construction of the source term is tested for the SIMPOL gas-aerosol system (Tsigaridis and Kanakidou, 2007) which describes partitioning of organic species.





## 2.11 Enabling Particle-Resolved GPU Chemistry

We integrated TChem-atm into the PartMC particle-resolved model to enable GPU-accelerated, chemically detailed simula-
tions of multiphase aerosol chemistry. PartMC is a stochastic, particle-resolved aerosol box model that resolves the composition
of many individual aerosol particles within a well-mixed volume of air. Riemer et al. (2009), DeVille et al. (2011), Curtis et al.
(2016), and DeVille et al. (2019) describe in detail the numerical methods used in PartMC. PartMC tracks a large number of
discrete computational particles to represent the aerosol population. Each computational particle is a vector of composition,
which tracks the mass of each species within the particle and evolves the composition in time. By coupling PartMC with
TChem-atm, each particle's chemistry is advanced using TChem-atm's batched source term evaluations and solver interface.
Kokkos ensures that these evaluations are offloaded to GPUs with minimal overhead.

This integration allows PartMC to take advantage of the performance portability and multiphase chemistry capabilities
developed in this work. The interface supports flexible mapping between PartMC's particle state variables and TChem-atm's
kinetic model structure, preserving modularity and extensibility. In test cases, we validated the consistency of results across
CPU and GPU architectures.

## 3 Results

This section presents two sets of results. First, we evaluate the chemical accuracy of the new TChem-atm integration within
PartMC by comparing its outputs against the established PartMC-CAMP model in a box model simulation (Section 3.1). This
includes gas-phase chemistry and gas–aerosol partitioning for a previously publised scenario (Dawson et al., 2022). Second,
we assess the computational performance of the TChem-based implementation across multiple hardware backends (CPUs and
GPUs) and solver configurations. We report per-particle wall-clock timings for both full ODE solves and individual kernel
components such as RHS and Jacobian evaluations. Together, these results demonstrate the correctness and scalability of
PartMC–TChem-atm for atmospheric chemistry modeling.

For this study, PartMC-TChem-atm was compiled and tested using Kokkos 4.5.01 with support for CPUs and GPUs. Simu-
lations were performed on the following computing architectures: an NVIDIA H100 GPU on DeltaAI at the National Center
for Supercomputing Applications (NCSA), University of Illinois; an AMD MI250x GPU on Frontier at the Oak Ridge Lead-
ership Computing Facility (OLCF); and an AMD EPYC 7763 CPU on Perlmutter at the National Energy Research Scientific
Computing Center (NERSC). The build process used the CMake-based infrastructure provided by TChem-atm, with Kokkos
execution and memory spaces selected at compile time to target the appropriate backend. GPU simulations used the CUDA and
HIP backends, while CPU runs used the OpenMP backend. All simulations used identical chemical mechanisms, numerical
tolerances, and time step settings to isolate hardware performance and consistency of results.





### 3.1 PartMC-TChem-atm box model setup and comparison with PartMC-CAMP

To evaluate the performance of TChem-atm, we conducted a box model simulation using PartMC. The model was configured to represent a well-mixed air parcel with constant temperature and pressure. This simulation focuses exclusively on gas-phase

chemistry and gas–aerosol partitioning, with coagulation and removal processes disabled. Gas-phase reactions were governed by the CB05 mechanism (Yarwood et al., 2005), and gas–aerosol partitioning followed the SIMPOL scheme (Tsigaridis and Kanakidou, 2007). The model was run for 24 hours with a time step of 60 seconds. Initial gas-phase concentrations and emissions were set according to Table 10 of Dawson et al. (2022), while the aerosol phase was initialized with three modes, as described in the same reference. For the simulations presented here, we initialized PartMC with $10^3$ computational particles.

Simulation results are shown in Figure 5. Panel (a) displays the 24-hour time series of ozone mixing ratios as predicted by PartMC-CAMP, PartMC-TChem-atm (CPU), and PartMC-TChem-atm (GPU). Panel (b)-(d) shows the evolution of gas-phase isoprene (ISOP) and its semi-volatile products, ISOP-P1 and ISOP-P2. ISOP-P1 is formed through the reaction of ISOP with OH, while ISOP-P2 results from its reaction with ozone. Figure 5(e)-(f) presents the aerosol-phase mass concentrations of ISOP-P1_aero and ISOP-P2_aero, which arise from the partitioning of their respective gas-phase precursors.

Per-timestep relative tolerances provided by TChem-atm to the CVODE solver control the final accumulated error. In order to achieve .01% global accuracy for this comparison, a relative tolerance of $10^{-6}$ was chosen. To summarize how PartMC-TChem-atm compares to PartMC-CAMP, the $\ell^2$-norm of the relative difference was $2.5 \times 10^{-5}$ for gas phase species and $8.2 \times 10^{-6}$ for aerosol species.

Overall, PartMC-TChem-atm reproduces the results of PartMC-CAMP with high fidelity. It performs consistently across

both CPU and GPU architectures, capturing expected chemical behavior while enabling accelerated simulations on heterogeneous computing platforms.

### 3.2 Benchmarking GPU Speedup and Solver Scaling

We evaluate the computational performance of TChem-atm on several hardware backends including multicore CPUs and both NVIDIA and AMD GPU architectures. To achieve this, the box model example from Section 3.1 was run by varying the

number of computational particles, which increases the size of the source term and the Jacobian matrix. We tested several ODE solver configurations presented in Table 2, including TINES TrBDF2 and two variants of Sundials CVODE that differ by the sparsity of the linear solver (Sundials CVODE-GMRES and Sundials CVODE (dense)). The same numerical parameters are applied for the three configurations.

For all numerical experiments, the model was run to a simulated time of 10 seconds and the wall clock time per time step

was measured. For each solver, the number of computational particles was varied from $N_{\mathrm{p}} = 1$ to $N_{\mathrm{p}} = 1 \cdot 10^6$ in twenty, logarithmically spaced increments. Figure 6 shows the wall clock time per particle for each solver configuration across a range of particle counts. For small particle numbers (i.e., $N_{\mathrm{p}} \lesssim 50$), solvers using dense linear algebra—namely `Sundials CVODE (dense)` and `TINES TrBDF2`—achieve the best performance due to their low overhead and efficient handling of





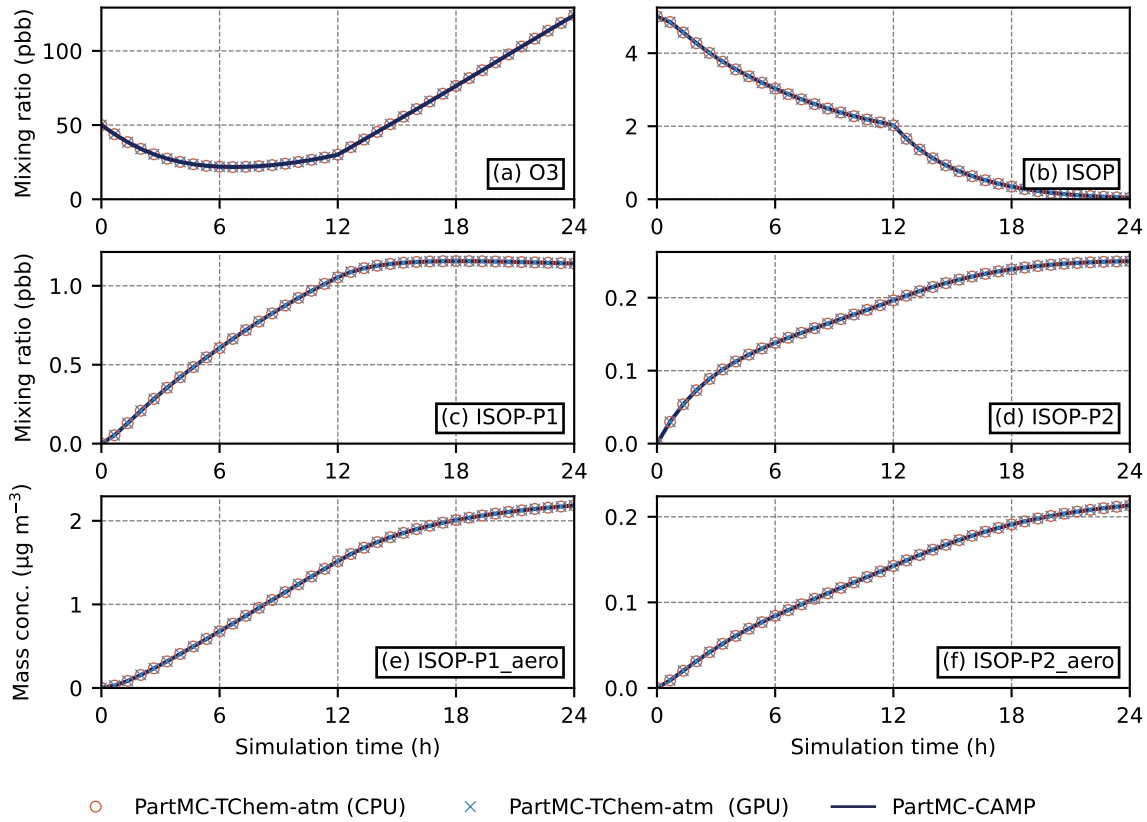

**Figure 5.** Comparison between PartMC-TChem-atm (CPU), PartMC-TChem-atm (GPU) and PartMC-CAMP for (a) ozone mixing ratio, (b) ISOP, (c) ISOP-P1 and (d) ISOP-P2 mixing ratios, (e) ISOP-P1_aero and (f) ISOP-P2_aero mass concentration for the 24-hour simulation period.

small Jacobians. However, as $N_p$ increases, the cost of forming and solving dense Jacobian systems grows rapidly, making
these approaches computationally prohibitive beyond $N_p \sim 500$.

In contrast, `Sundials CVODE-GMRES` employs a Jacobian-free Krylov method, which avoids explicit construction of the Jacobian. This enables continued scaling to larger particle numbers, with wall clock time per particle decreasing even up to $N_p = 10^6$ on GPU architectures. These results underscore the importance of matching solver structure to problem size and hardware—dense solvers excel at small scale, while Jacobian-free methods offer superior scalability.

In the GMRES case, we observe substantial speedups of GPU relative to CPU at large computational particle numbers, underscoring the advantages of Jacobian-free Krylov methods on GPUs. These results are based on a single-cell configuration, however, and therefore do not yet fully exploit the parallelism available on GPU architectures. In larger multi-cell simulations, we anticipate even greater gains for GMRES as well as improvements for dense solvers, which are not fully reflected under the present single-cell conditions.




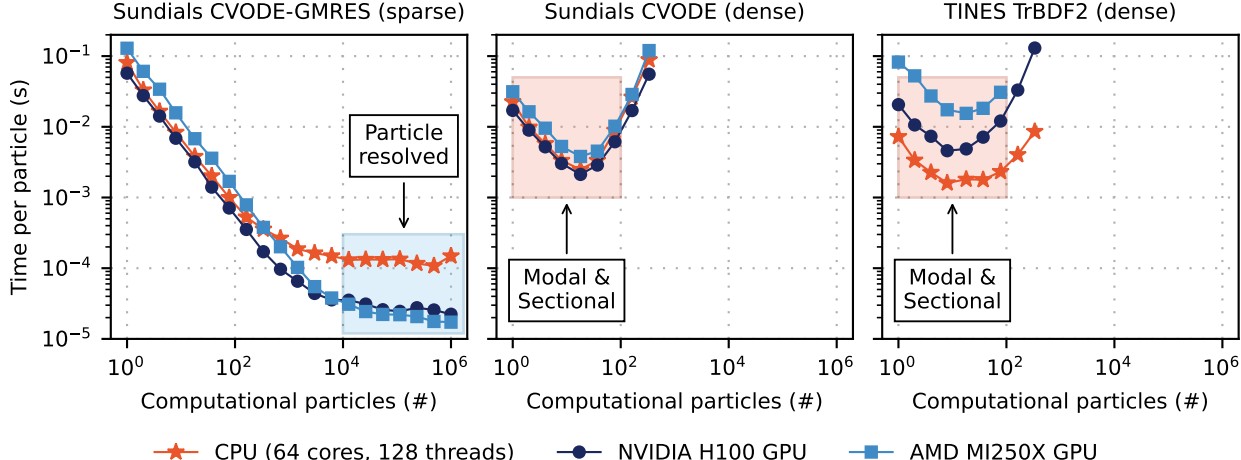

**Figure 6.** Per-particle solver wall clock time through $t = 10$ s organized by solver. Timings are colored by computing architecture. For each solver and architecture, team size and vector size were set via the `Kokkos::AUTO` configuration. `TINES TrBDF2` timings for the AMD MI250X at particle counts larger than $N_\mathrm{p} \sim 100$ are not included due to the required memory exceeding allocatable GPU DRAM capacity. Shaded regions indicate approximate computing regimes for each aerosol treatment.

**Table 2.** Numerical parameters for evaluated ODE solvers.

| Parameter | Sundials CVODE-GMRES | Sundials CVODE (dense) | TINES TrBDF2 |
|---|---|---|---|
| Minimum time step | 1 | 1 | $1 \times 10^{-3}$ |
| Maximum time step | 1 | 1 | 1 |
| Absolute tolerance | $1 \times 10^{-16}$ | $1 \times 10^{-16}$ | $1 \times 10^{-16}$ |
| Relative tolerance | $1 \times 10^{-8}$ | $1 \times 10^{-8}$ | $1 \times 10^{-8}$ |
| Linear solver type | GMRES | LU Dense | UTV dense |

These results have practical implications for model selection and hardware deployment. Particle-resolved simulations such as PartMC typically require on the order of 10,000 computational particles per grid box to adequately resolve mixing state and aerosol composition. In this regime, dense linear solvers become inefficient, and the GPU implementation of Sundials CVODE-GMRES offers significant performance advantages due to its Jacobian-free formulation and better scalability with particle number.

Conversely, if the application involves simplified aerosol representations—such as modal or sectional models with a small number of size/composition bins—the computational burden is much lower, and CPU-based solvers with dense linear algebra may offer better performance. Thus, the optimal hardware–solver combination depends strongly on the complexity and resolution of the aerosol representation. To further illustrate the relationship between aerosol treatment and the choice of solver, approximate computing regimes for each aerosol treatment are shown as shaded regions in Figure 6.





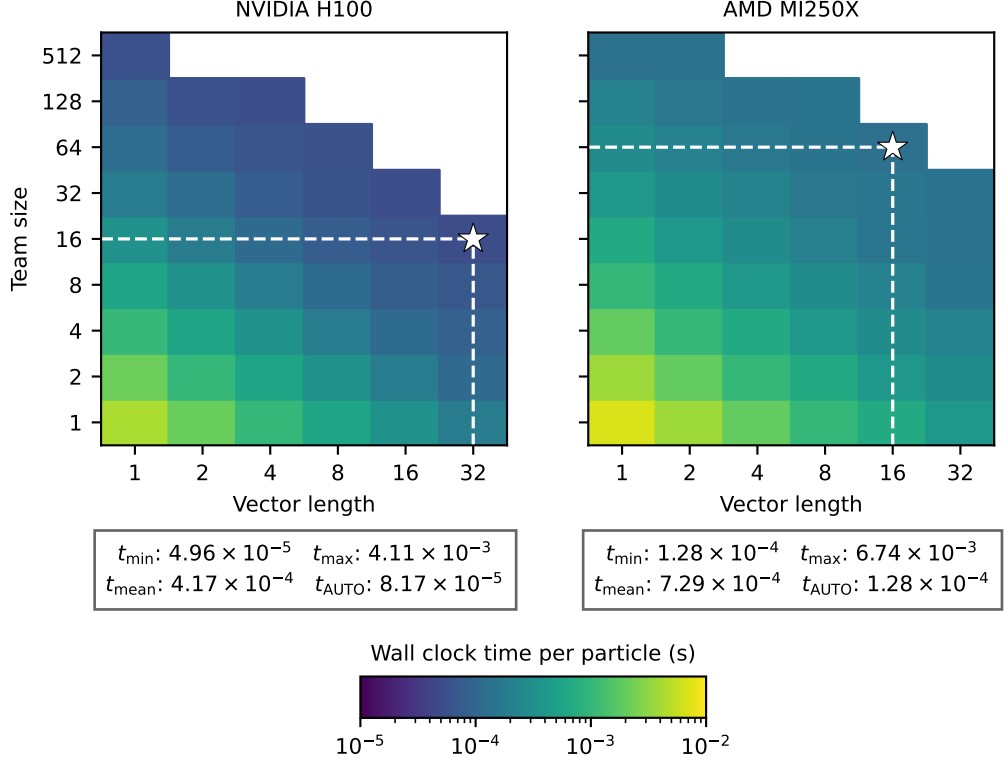

**Figure 7.** Heatmaps of wall clock time per particle as a function of Kokkos team size and vector length for the Sundials CVODE-GMRES solver on the NVIDIA H100 (left) and AMD MI250X (right). For each solver run, the number of particles was fixed at 1000. The optimal configuration (lowest wall clock time) is indicated by a white star in each plot.

Timings shown in Figure 6 were conducted using the `Kokkos:AUTO` configuration, which automatically sets the level of parallelism at both the team and vector level (for those familiar with NVIDIA systems, the Kokkos team size and vector size are cast as the $x$ and $y$ dimension of each CUDA block). Both parameters can be set at runtime to determine a problem-specific optimal configuration for team and vector size. Figure 7 shows heatmaps of wall clock time per particle for the Sundials CVODE-GMRES solver where both the team size and vector size are varied. Optimal team and vector size configurations

differ by GPU. For the NVIDIA H100, a team size of 16 and a vector length of 32 resulted in the lowest wall clock time per particle, whereas a team size of 64 and vector length of 16 were best for the AMD MI250X. Below each heatmap in Figure 7 are the minimum, maximum, and mean wall clock time across each team and vector configuration alongside the wall clock time achieved by the `Kokkos:AUTO` configuration. We find that the optimal configuration for the NVIDIA H100 is within a factor of 2 faster than the `Kokkos:AUTO` configuration, while both are equal for the AMD MI250X.

Ultimately, solver performance is governed by the size and structure of the Jacobian matrix. A large number of particles or reactions leads to a large Jacobian, while a large number of chemical species often results in a sparse one. The balance





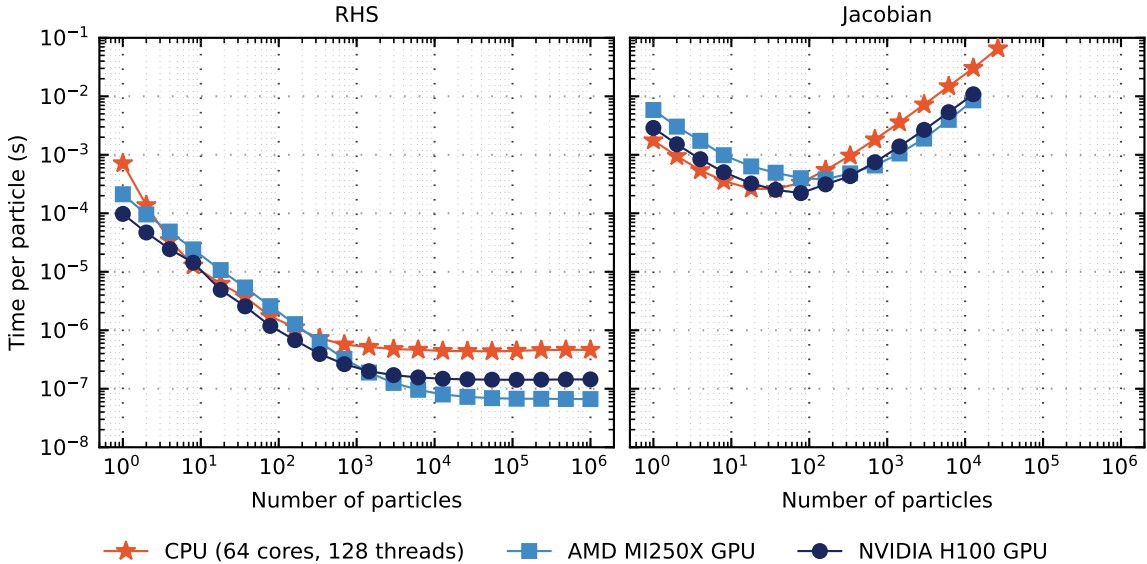

**Figure 8.** Per-particle wall clock time for evaluation of the gas-aerosol right-hand side (left) and Jacobian (right). Timings are colored by computing architecture. For each timing run, team size and vector size were set via the `Kokkos::AUTO` configuration. Particle number is limited for Jacobian evaluations due to exceeding the allocatable memory of each computing architecture.

between these factors—reaction count, species count, and particle/bin resolution—ultimately determines whether dense or iterative solvers, and CPU or GPU hardware, offer the most efficient path forward.

In addition to full ODE solver runs, a separate set of timings were conducted to evaluate the performance of the RHS
evaluation and Jacobian construction. Figure 8 shows how wall clock time for the RHS and Jacobian vary as the number of particles increases.

For each computing backend, per-particle performance for RHS evaluations improves as the number of particles increases. When $N_\mathrm{p} \lesssim 1 \cdot 10^3$, we find the best performance for the NVIDIA H100. Above $N_\mathrm{p} = 1 \cdot 10^3$, the AMD MI250X achieves the best performance through $N_\mathrm{p} = 1 \cdot 10^6$. At high particle counts ($N_\mathrm{p} \gtrsim 1 \cdot 10^4$), performance plateaus for each computing
backend. At $N_\mathrm{p} = 1 \cdot 10^6$, GPU backends achieve speedups of 2.3 and 5 relative to the CPU for the NVIDIA H100 and AMD MI250X, respectively.

Additionally, Jacobian evaluation timings exhibit similar performance for both the CPU and GPUs: wall clock time per particle decreases when the number of particles is low (i.e., $N_\mathrm{p} \lesssim 100$), however, performance degrades as $N_\mathrm{p}$ increases further due to corresponding growth in the Jacobian.



## 4 Conclusion

We have presented TChem-atm, a performance-portable approach for simulating chemically detailed, multiphase atmospheric chemistry across diverse computing architectures. By integrating the chemical mechanism infrastructure of CAMP with the Kokkos-based backend of TChem-atm, TChem-atm enables consistent deployment across CPUs and GPUs while maintaining flexibility in mechanism configuration and solver selection.

A key strength of TChem-atm is its support for multiple aerosol representations—including modal, sectional, and particle-resolved approaches—making it suitable for a wide range of host models and applications. Integration with the particle-resolved model PartMC demonstrates that TChem-atm accurately reproduces gas-phase chemistry and gas–aerosol partitioning results, verifying the method's correctness and interoperability.

Performance benchmarks reveal that dense linear solvers are well-suited for small particle populations, whereas Jacobian-free solvers such as CVODE-GMRES offer scalable performance benefits on GPUs as the number of computational particles increases. These gains are particularly significant for particle-resolved models like PartMC, which typically require around 10,000 particles per grid cell to capture aerosol mixing state. Across solvers, our modular approach supports efficient, scalable integration of detailed multiphase chemistry across diverse computing platforms.

While the current version of TChem-atm is not yet fully performance-optimized, it provides a solid and extensible foundation for future development. Ongoing efforts will focus on enhanced solver strategies, improved memory layout, and broader chemical mechanism support. The code and all associated test cases are publicly available under an open-source license, enabling verification, reuse, and extension by the broader atmospheric modeling community.

*Code and data availability.* TChem-atm is open source and available via GitHub under the BSD 2-Clause License at: https://github.com/pcLAeroParams/TChem-atm/. The version of the code used in this study has been archived on Zenodo under https://doi.org/10.5281/zenodo.17058144 (Diaz-Ibarra et al., 2025a). TChem-atm depends on Kokkos, SUNDIALS, and Tines, which are also openly available. PartMC is open source and available via GitHub at: https://github.com/compdyn/PartMC. All simulation inputs and output used to generate the figures in this paper are archived at the Illinois Data Bank under https://doi.org/10.13012/B2IDB-3697767_V1 (Diaz-Ibarra et al., 2025b).

*Author contributions.* OD led the implementation and development of TChem-atm, coordinated computing resources. SGF developed code and performed benchmarking and scalability analysis (Section 3.2), curated data, coordinated computing resources. JHC led the integration of TChem-atm and PartMC, conducted simulations and analysis (Section 3.1) and curated data. ZD performed code review and technical validation. MW provided overall guidance. NR coordinated the project, outlined and organized the manuscript, and synthesized the final text. All authors contributed to editing.

*Competing interests.* The authors declare that they have no conflict of interest.



*Acknowledgements.* This work used computing resources from DeltaAI at the National Center for Supercomputing Applications (NCSA),
University of Illinois Urbana-Champaign; Frontier at the Oak Ridge Leadership Computing Facility (OLCF); and Perlmutter at the National
Energy Research Scientific Computing Center (NERSC). The authors thank the developers of Kokkos, TChem, Tines, and SUNDIALS for
their contributions to the open-source ecosystem that made this work possible.

This work was supported by the U.S. Department of Energy, Office of Science, Office of Biological and Environmental Research under
Award Number DE-SC0022130, the National Science Foundation Grant No. AGS 19-41110, and the Laboratory Directed Research and
Development program at Sandia National Laboratories. Sandia National Laboratories is a multimission laboratory managed and operated
by National Technology and Engineering Solutions of Sandia LLC, a wholly owned subsidiary of Honeywell International Inc. for the U.S.
Department of Energy's National Nuclear Security Administration contract DE-NA0003525.

The employees co-own right, title and interest in and to the article and are responsible for its contents. The United States Government
retains, and the publisher, by accepting the article for publication, acknowledges that the United States Government retains a non-exclusive,
paid-up, irrevocable, world-wide license to publish or reproduce the published form of this article or allow others to do so, for United States
Government purposes. The DOE will provide public access to these results of federally sponsored research in accordance with the DOE
Public Access Plan at https://www.energy.gov/downloads/doe-public-access-plan.



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
