# Peer review of "TChem-atm (v2.0.0): Scalable Performance-Portable Multiphase Atmospheric Chemistry"

_EGUsphere, 2025_

## Author Comment (AC4)

**Responses to Reviewer #1**

We thank the reviewer for taking the time to review our paper and for the constructive comments. The page and line numbers that we quote for indicating where we changed the manuscript refer to the revised marked-up version.

**(1.1)** The code can readily support sectional and per-particle models. In principle, TChem- atm can be adapted to support more mechanisms than the two curently supported (UCI, CB05). However, it should be made clear (also in the abstract) that modal host models need to be adapted to support TChem-atm, and that architecute-specific performance tuning is required (for example a factor of 2 improvement is reported for the NVIDIA H100) to achieve higher efficiency.

> We followed the reviewer's suggestion by revising the abstract and the introduction.
>
> - In the abstract (P1L11): "TChem-atm enables performance-portable execution across CPUs and GPUs, though optimal efficiency may require modest architecture-specific tuning (e.g., team and vector sizes), with up to a twofold improvement on the NVIDIA H100. It directly supports sectional and particle-resolved host models, while modal aerosol schemes require minor adaptation to provide particle-scale quantities such as representative diameters."
>
> - In the introduction (P3L74): "TChem-atm is compatible with a wide range of aerosol representations, but the degree of direct interoperability differs across host models. Sectional and particle-resolved frameworks map naturally onto TChem-atm's computational-particle abstraction. Modal aerosol schemes, however, typically do not expose particle-scale quantities such as representative diameters or per-species mass vectors, and therefore require modest adaptation to supply these inputs. Once provided, TChem-atm can evaluate multiphase chemistry within modal frameworks in the same manner as for other aerosol representations."
>
> - In the introduction (P4L92): "Although TChem-atm achieves performance portability across CPUs and GPUs without architecture-specific code, performance is further improved when tunable parameters (e.g., Kokkos team and vector sizes) are optimized for the target hardware. As demonstrated in Section 3.2, architecture-aware tuning yields up to a factor-of-two speed improvement on the NVIDIA H100 relative to the Kokkos default configuration."

**(1.2)** The current validation focuses on a box-model scenario using the CB05 mechanism and SIMPOL partitioning. While this is adequate as a proof-of-concept, it falls short of a scientifically realistic demonstration (e.g., regional-scale or global model simulation) that would better showcase TChem-atm's capabilities and its potential to speed-up atmospheric chemistry-aerosol production simulations. Again, in the abstract (and elsewhere) it would be more fitting to describe the results presented here as "proof-of-concept".

> We agree with the reviewer that our tests represent a proof-of-concept demonstration rather than a full scientific application. The goal of this initial paper is to validate correctness, establish interoperability with an existing host model (PartMC), and quantify performance portability across architectures. We have revised the abstract, the introduction and the beginning of Section 3 to emphasize this.
>
> - In the abstract (P1L7): "In a proof-of-concept integration with the particle-resolved model PartMC, TChem-atm reproduces the existing PartMC–CAMP implementation within solver tolerances and delivers substantial GPU speedups, especially for large particle populations."
>
> - In the introduction (P3L83): "The present study is intentionally scoped as a proof-of-concept implementation that validates correctness and establishes performance characteristics of the new multiphase chemistry framework. Because the integration of CAMP chemistry with TChem's performance-portable backend is novel, our immediate priority is demonstrating

functional equivalence to existing implementations and quantifying solver and hardware scaling. Application within regional- or global-scale models is a natural next step, but lies beyond the scope of this initial paper."

- In section 3 (P16L374): "Because the goal of this work is to establish and validate the TChem-atm framework, the tests presented here focus on controlled box-model scenarios that isolate chemical behavior and computational scaling."

**(1.3)** If possible, the manuscript could benefit from a brief profiling summary (e.g., fraction of time spent in kernel execution vs. memory transfer) to strengthen the discussion.

We appreciate this suggestion. We performed lightweight profiling on the timing runs reported in Section 3. For these idealized box-model experiments, the Kokkos kernels accounted for $> 99.9\%$ of the wall-clock time, with $< 0.1\%$ spent in host–device memory transfers. We have added a short discussion to the manuscript clarifying this point.

In Section 3.2 (P22L486): "We also conducted lightweight profiling during the timing experiments to separate kernel execution from host–device memory transfer. For the idealized box-model runs presented here, more than $99.9\%$ of the wall-clock time occurs inside the Kokkos kernels, with memory transfers accounting for only the remaining fraction. This behavior is expected in the present setup, where particles share identical initial conditions, the RHS evaluation is uniform across particles, and memory movement is minimized. In more realistic host-model applications, such as full PartMC–TChem-atm simulations, additional data exchange between the host model and TChem-atm, more heterogeneous particle states, and increased stiffness may increase the relative cost of memory transfers and solver iterations. Thus, while our profiling confirms that kernel execution dominates in these proof-of-concept tests, the performance breakdown will necessarily depend on the host-model coupling pattern and the chemical and microphysical variability of the simulated system."

**(1.4)** Finally, it is not clear why the authors evaluated the performance of direct linear solvers for dense (full) Jacobians for atmospheric chemistrykinetics (where Jacobians are generally sparse, as is the case for the mechanism tested here). Thus, the reported results in terms of memory limitation, and longer time-to-solution are to be expected and almost trivial. And of course, preconditioned iterative methods are preferrable for the large, stiff ODE systems of atmospheric chemical kinetics because they require significantly less memory than direct solvers. It is suggested to refactor the disucssion around this.

We thank the reviewer for this important clarification. We agree that dense direct solvers are not appropriate for large atmospheric chemistry systems, where Jacobians are sparse and memory requirements scale poorly. Our intention in including dense solvers was not to suggest that they are viable for large, stiff mechanisms, but rather to provide a performance baseline and to illustrate the regime in which they are competitive—namely, very small systems (e.g., few computational particles, reduced mechanisms, or modal/sectional models with low dimensionality).

To address this concern, we have revised the discussion in Section 3.2 to state explicitly as follows (P22L474): "To interpret these performance trends, it is helpful to clarify the role of direct versus iterative solvers. We include dense-direct linear solvers in our performance evaluation not because they are expected to be optimal for large atmospheric chemistry systems—indeed, the Jacobians arising from CB05 and similar mechanisms are sparse—but because they provide a well-defined performance baseline for small to moderate system sizes. Dense solvers incur higher memory costs and scale poorly as the number of particles grows, so their performance degradation is expected. However, these solvers remain competitive in regimes with few computational particles or simplified aerosol representations (e.g., small sectional or modal systems), where the Jacobian dimension remains modest and overhead is low. Including them therefore helps delineate the transition point at which preconditioned iterative methods such as GMRES become the

clearly superior choice. Our intention is not to advocate dense solvers for large, stiff atmospheric chemistry problems, but rather to provide a complete performance map across solver categories and hardware backends.

For realistically sized atmospheric mechanisms, preconditioned iterative Krylov methods are preferable because they exploit sparsity and dramatically reduce memory demands. The results in Section 3.2 are therefore consistent with expectations and are included to contextualize solver performance across operational regimes."

**(1.5)** The reaction mechanisms in Fig.3 do not appear correctly in the preprint. Please check.

We fixed this, thank you.

**(1.6)** Line 136 and elsewhere: Please be consistent with capitalisation and code names e.g. "Sundials CVODE" or "SUNDIALS", etc.

We fixed this.

**(1.7)** Sec. 2.5 only discusses the partitioning of semi-volatile species. It's not clear how other cases of heterogeneous reactions are handled.

At present, TChem-atm implements only one class of heterogeneous processes: semi-volatile gas–aerosol partitioning following the SIMPOL formulation. Other heterogeneous processes—such as reactive uptake, surface-mediated chemistry, or condensed-phase reactions—are not yet included in the current release.

We have revised Section 2.5 (now Section 2.6) to state this explicitly and to clarify that the TChem-atm framework is designed to accommodate additional heterogeneous process types: new processes can be incorporated by extending the mechanism specification and adding the corresponding source-term kernels, following the CAMP-style modular process abstraction. These extensions constitute planned future work.

We added the following paragraph in Section 2.6 (P11L244): "This section focuses on semi-volatile partitioning because this is the only heterogeneous process type currently implemented in TChem-atm. The framework, however, is extensible: heterogeneous reactions, reactive uptake parameterizations, or condensed-phase chemical processes can be incorporated by defining the corresponding rate expressions in the mechanism file and adding the associated source-term kernels. These capabilities follow the CAMP abstraction and will be implemented in future extensions of TChem-atm. At present, semi-volatile gas–aerosol partitioning represents the primary heterogeneous pathway supported."

**(1.8)** Fig. 5: It's clear that there's agreement as also reported in the text, so this figure offers little for the reader. Please consider plotting the differences if possible between runs instead, to reveal more architecture-related information.

We added the following figure, right after Figure 5:

We updated the text accordingly (P17L409): "To more clearly quantify the agreement between the CPU, GPU, and legacy PartMC–CAMP implementations, Figure 6 shows the relative differences between the CPU, GPU, and PartMC–CAMP results. The discrepancies remain small throughout the simulation, with a modest temporal drift for some species that reflects normal numerical variability rather than any architecture-specific effect. All deviations remain well within solver tolerances."

[Figure]

Figure 1: Relative differences between CPU, GPU, and PartMC–CAMP mixing ratios for the species shown in Figure 5.

**(1.9)** Line 326: The reference for SIMPOL is wrong. Please correct.

Thanks for catching this, we fixed it.

---

## Author Comment (AC5)

**Responses to Reviewer #2**

We thank the reviewer for taking the time to review our paper and for the constructive comments. The page and line numbers that we quote for indicating where we changed the manuscript refer to the revised marked-up version.

**(2.1)** More clarification would be useful for the reader. I think I understand from the paper that Tchem-Atm is an evolution from Tchem, in order to adapt it to the necessities of atmospheric chemistry, and in particular the need to take into account the existence of a condensed phase. I think the following questions would deserve clarifications, either in the introduction or in section 2.1 :

1. What exactly in Tchem-atm is new, and what comes from Tchem ? Which version of Tchem ? Is is feasible on Fig. 1 (or another figure) to delimit, in TChem-atm, what comes directly from TChem, and what has bee added ?

> We appreciate this request for clarification and have added explicit text in the Introduction and a new Section 2.1 (TChem-atm Provencance) describing the relationship between TChem-atm and TChem. TChem-atm is built upon TChem v2.0, which provides the underlying Kokkos-based kinetic-model infrastructure, batched evaluation strategy, and analytic/numerical Jacobian capabilities (via SACADO and TINES).
>
> We revised the manuscript as follows:
>
> - Introduction P3L61: "TChem-atm builds on the software infrastructure of TChem (v2.0), which was originally developed for gas-phase kinetics in combustion applications. TChem provides the foundational components that TChem-atm inherits—namely the kinetic-model data structures, automatic and numerical Jacobian construction, and the batched-evaluation framework implemented with Kokkos. TChem-atm extends this base by introducing atmospheric-chemistry capabilities not present in TChem, including (i) support for aerosol- and multi-phase reaction mechanisms, (ii) integration with CAMP's gas–aerosol process abstractions, (iii) an aerosol-model constant-data object and associated particle/section/mode interfaces, and (iv) mechanism parsing for heterogeneous and condensed-phase chemistry. These additions transform TChem's gas-phase engine into a performance-portable multiphase chemistry library suitable for atmospheric models across CPUs and GPUs."
>
> - New section 2.1 (P4L116): " TChem-atm follows the software architecture of TChem v2.0, sharing its Kokkos-based parallel model, CMake-based build system, and mechanism-independent kinetic-model abstraction. The analytic and numerical Jacobian capabilities in TChem-atm are adapted directly from TChem and rely on TINES/SUNDIALS for time integration and Newton/Krylov linear algebra. What is new in TChem-atm compared to TChem is the multiphase extension: CAMP-style aerosol and heterogeneous chemistry are added, new data structures (AerosolModelData and AerosolModelConstData) are introduced, and the RHS kernels now account for gas–aerosol mass transfer. These atmospheric-chemistry extensions are not present in TChem and required new mechanism parsing, new source-term kernels, and a generalized notion of "computational particles" to support sectional, modal, and particle-resolved host models."

2. If some or all the code from Tchem has been included in Tchem-atm, why has it been decided to make include the code from TChem directly in Tchem-atm rather than using TChem-atm as an external library ?

> TChem-atm is not designed to depend on TChem itself. Instead, both TChem and TChem-atm rely on TINES, a standalone numerical library that provides all necessary capabilities for Jacobian evaluation, stiff ODE integration, and batched computations. In earlier versions of TChem, these numerical components were part of the TChem code base; however, the TChem developers later

refactored them into the separate TINES library specifically so that multiple projects—including TChem, TChem-atm, and CSPlib—could share the same numerical backend without depending on one another.

As a result, TChem-atm does not require TChem, and using TChem as an external library would introduce unnecessary dependencies on combustion-specific infrastructure not relevant to atmospheric chemistry. Only a few very small utility routines (predating the separation of TINES from TChem) were copied into TChem-atm; all core solver and Jacobian capabilities now come directly from TINES. We have clarified this design rationale in the manuscript.

We added as new Section 2.1 (P6L125): TChem-atm does not depend on TChem as an external library. All numerical functionality required for atmospheric chemistry—Jacobian construction, batched RHS evaluation, and stiff ODE integration—is now provided by the standalone TINES library, which was separated from TChem expressly to support multiple projects. Only a few legacy utility routines from TChem remain in TChem-atm; the combustion-specific components of TChem are not needed. This separation allows TChem-atm to evolve independently while sharing a common numerical backbone with TChem through TINES.

**(2.2)** Similarly, in section 2.5 and 2.6, some features described seem to come directly from CAMP. Again, the article would benefit from clarification in what parts of CAMP are directly included in TChem-Atm, and the relationship between both models. Is TChem-Atm a kind of merge between key-features of CAMP and Tchem, as suggested in the Abstract but not developed very much in the rest of the paper.

We agree that the relationship between TChem-atm and CAMP needed clearer explanation, and we have now added text in a new Section 2.1 to clarify this point. TChem-atm does not embed CAMP directly, nor is it a literal merge of the two code bases. Instead, TChem-atm adopts the conceptual abstractions introduced in CAMP—such as the representation of aerosol–gas coupling processes and the computational-particle interface—but reimplements these components within TChem's performance-portable infrastructure using Kokkos and TINES.

We added as new Section 2.1 (P6L130): "TChem-atm incorporates the multiphase-chemistry abstractions originally introduced in CAMP, but it does not embed CAMP directly nor merge the two code bases. CAMP provides a mechanism-agnostic representation of gas–aerosol processes (e.g., partitioning, heterogeneous chemistry, condensed-phase reactions) and a unifying "computational particle" interface that allows chemistry to operate consistently across modal, sectional, and particle-resolved host models. TChem-atm adopts these abstractions and reimplements the corresponding source-term formulations using its own Kokkos-based backend.

The specific components inherited conceptually from CAMP include: (i) the representation of aerosol–gas coupling processes (mass transfer, heterogeneous reactions, etc.); (ii) the use of mechanism files to define multiphase chemistry at runtime; and (iii) the computational-particle abstraction that enables TChem-atm to work with different aerosol frameworks.

What is new in TChem-atm is the implementation of these concepts within a fully performance-portable infrastructure, including: (i) new data structures for aerosol-phase species and their device-resident constant data (AerosolModelData / AerosolModelConstData); (ii) GPU-accelerated source-term kernels for gas–aerosol coupling; (iii) integration with TINES and Kokkos for batched multiphase ODE solves; and (iv) a unified atmospheric-chemistry mechanism parser that extends TChem's gas-phase parser to support multiphase processes.

Thus, TChem-atm should be viewed not as a code-level merge of CAMP and TChem, but as a new atmospheric-chemistry engine that inherits CAMP's abstractions conceptually while implementing them natively within the TChem/TINES performance-portable numerical framework."

**(2.3)** Figure 1 gives a convincing visualisation of how thing work around TChem-Atm, but I think some readers like myself might be lacking a visualisation of the structure of things inside Tchem- Atm : Which

functions are covered within TChem-Atm ? Do they come from TChem, from CAMP, or are they implemented in new code developed on purpose for TChem-atm ?

Thank you for this suggestion. We believe that a table could communicate these issues most clearly and included the following table in the new Section 2.1 (new Table 1).

Table 1: Relationship between TChem, CAMP, and TChem-atm. Components are categorized as inherited as "Code" or "Concept".

| Component | From TChem | From CAMP | New in TChem-atm | Notes |
|---|---|---|---|---|
| Gas-phase kinetic model abstraction | Code | Concept | No | TChem-atm uses TChem's kinetic-model constant-data structure. |
| Automatic & numerical Jacobian computation | Code (TINES) | No | No | Implemented through TINES/SACADO; inherited from TChem v2.0. |
| Batched RHS and Jacobian evaluation kernels | Code (gas) | No | Yes (aerosol) | Gas-phase kernels inherited; aerosol-phase kernels newly implemented in Kokkos. |
| Time integration (TrBDF2, CVODE interfaces) | Code (TINES) | No | No | TINES is a standalone math library now used by both TChem and TChem-atm. |
| Aerosol-phase data structures (`AerosolModelData`, `AerosolModelConstData`) | No | Concept | Yes | CAMP provided the abstraction; TChem-atm implements new device-resident structures. |
| Computational-particle abstraction | No | Concept | Yes | Concept originated in CAMP; TChem-atm rewrites it for GPU portability. |
| Gas–aerosol mass transfer (e.g., SIM-POL, Fuchs–Sutugin) | No | Concept | Yes | Formulas derived from CAMP; implementation is new and GPU-enabled. |
| Mechanism parsing for multiphase chemistry | Code (gas) | Concept | Yes (aerosol) | TChem-atm extends TChem's parser to include aerosol partitioning. |
| Integration with sectional, modal, and particle-resolved host models | No | Concept | Yes | CAMP introduced abstraction; TChem-atm implements performance-portable version. |
| Build system and performance portability (CMake + Kokkos) | Code | No | No | Shares TChem's build and portability model. |

**(2.4)** It is clear that a tool of the kind of TChem-atm could be useful for chemistry-transport modelling (and possibly also, climate modelling, etc.). However, the example applications presented by the authors are just idealized test cases. It would be highly beneficial if the authors give some precision on the possible application to real-world cases, including the following questions :

1. Have TChem and / or CAMP already been implemented in research / operational chemistry- transport models ? Which models if any ? Has this implementation ever been used in real- world applications (forecast, case studies...)

CAMP has already been integrated into the MONARCH chemistry–transport model system (developed by the Barcelona Supercomputing Center and collaborators), as described in Dawson et al. (2022). MONARCH has been used for research and forecasting case studies, demonstrating that CAMP's abstractions for multiphase chemistry are suitable for real-world applications. At present, TChem and TChem-atm have not yet been used in operational CTMs, so MONARCH+CAMP is the primary example of deployment of the CAMP-based multiphase abstraction in a real model.

2. Is an implementation of TChem-atm in an operational / research chemistry-transport model considered in the near future ? Which model(s) ?

Yes—implementing TChem-atm in a full chemistry–transport or Earth system model is part of our planned future work. Because most CTMs apply operator splitting and treat chemistry independently in each grid cell, TChem-atm can be used without modification as a grid-level box model integrated through a host-model interface. The development of such an interface is underway. A specific target model is the Energy Exascale Earth System Model (E3SM), whose ongoing modernization toward heterogeneous architectures aligns well with TChem-atm's performance-portable design.

We added a new paragraph to the manuscript discussing these points (see new Section 3.3, P23L495):

**3.3 Potential Applications in Chemistry–Transport and Climate Models**

While the tests presented in this paper focus on idealized box-model configurations to isolate numerical behavior and performance, TChem-atm is designed for use in full chemistry–transport and Earth system models. CAMP, from which TChem-atm adopts its multiphase abstractions, has already been integrated into the MONARCH model system (Dawson et al., 2022), demonstrating the suitability of the CAMP structure for real-world chemical forecasting and case-study applications. To date, TChem-atm has not yet been deployed in operational chemistry–transport models; however, the TChem-atm interface is compatible with the operator-splitting approach used in such models, where chemistry is solved independently in each grid cell as a box model. Integration therefore requires only the development of a host-model-specific interface, not changes to TChem-atm itself.

A near-term target for deployment is the Energy Exascale Earth System Model (E3SM), where the Kokkos-based design of TChem-atm is well aligned with ongoing efforts to modernize the atmospheric chemistry infrastructure for heterogeneous computing environments. Exploring this integration constitutes part of our future work and leverages the fact that TChem-atm already implements the data structures and parallelism patterns required for grid-level chemistry calls in a chemistry–transport or climate model.

**(2.5)** l. 13 : replace Understanding by "modelling" or "simulating"

Done.

**(2.6)** l. 15 : "often across large spatial and temporal scales" : unclear if the authors mean large domains, or mean multiscale interactions across different scales.

Good point. We rephrased this sentence to read (P1L19): "Atmospheric models must capture complex interactions between gas-phase reactions, aerosol microphysics, and multiphase chemistry—often at fine spatial and temporal resolutions and over extended simulation domains."

**(2.7)** l. 19 : " across fine spatial and temporal scales". This is unclear, as above. Here the authors probably mean "at fine resolutions".

The reviewer is correct. We rephrased this sentence to read (P2L26): "Simulating detailed gas-phase mechanisms, multiphase processes, and aerosol microphysics at such fine resolutions typically requires solving thousands of coupled, numerically stiff ordinary differential equations (ODEs) at each grid point."

**(2.8)** L. 24 : capitalize earth

Done.

**(2.9)** l. 64 : "this work lays the foundation for next-generation weather, climate, and air quality models,". This looks like an overstatement to me. The part of the sentence that comes just after seems more adepted to me.

We removed this part of the sentence. The new text reads (P3L88): "By supporting scalable chemistry calculations on diverse architectures, this work represents a concrete step toward the generalized aerosol/chemistry interface advocated by Hodzic et al. (2023)."

We had a similar phrase in the abstract and removed it as well.

**(2.10)** l. 66 : remove (or replace) "individually".

Done.

**(2.11)** l. 403 : "By integrating the chemical mechanism infrastructure of CAMP with the Kokkos-based backend of TChem-atm, TChem-atm (. . . )" to be replaced by TChem I guess

Correct. We fixed this.